# Analysis of Muscle Activity Following the Application of Myofascial Release Techniques for Low-Back Pain—A Randomized-Controlled Trial

**DOI:** 10.3390/jcm10184039

**Published:** 2021-09-07

**Authors:** Piotr Ożóg, Magdalena Weber-Rajek, Agnieszka Radzimińska, Aleksander Goch

**Affiliations:** Department of Physiotherapy, Collegium Medicum in Bydgoszcz, Nicolaus Copernicus University, 85-067 Bydgoszcz, Poland; m.weber@cm.umk.pl (M.W.-R.); radziminska@cm.umk.pl (A.R.); katfizjoter@cm.umk.pl (A.G.)

**Keywords:** muscle activity, low-back pain, myofascial release

## Abstract

Introduction. Lumbosacral dysfunctions and the resulting pain syndromes, such as low-back pain (LBP), are one of the most common musculoskeletal problems being faced by society around the world. So far, a contributory role of thoracolumbar fascia (TLF) dysfunction in some cases of LBP has been suggested. Research also confirms that muscle resting activity level in the TLF area is increased in people with LBP. Myofascial release (MFR) is a therapeutic option offered to patients with chronic low-back pain (CLBP). The therapy aims to improve flexibility and sliding between layers of soft tissue, and thus decrease muscle activity, reduce pain intensity, and improve functional performance. Objective. This study aims to assess changes in resting activity of selected muscles within the TLF in a group of patients with CLBP immediately after a single MFR treatment and one month after the intervention. Methods. A total of 113 patients with CLBP completed the study. Simple randomization was applied to assign subjects to study groups. The experimental group (*n* = 59) underwent a single session of MFR therapy. No therapeutic intervention was applied to the control group (*n* = 54). Surface electromyography was used to evaluate positive treatment effects in patients immediately after receiving the therapy (experimental group) and after one month (experimental and control group). Results. A statistically reliable decrease in the activity of erector spinae (ES) and multifidus muscles (MF) was observed after a single session of MFR therapy. Effects of the treatment were present immediately after receiving the therapy and one month after the intervention. Conclusions. A single MFR treatment in patients with CLBP immediately reduces the resting activity levels of ES and MF. Results of measurements carried out one month after the treatment confirm that the therapeutic effects were maintained.

## 1. Introduction

Lumbosacral dysfunctions and the resulting pain syndromes, such as LBP, are one of the most common musculoskeletal problems that affect our society. Chronic low-back pain is problematic for patients, because the condition limits their physical fitness for a long time, and it is diagnostically challenging for physicians due to the condition’s complex multifactorial etiology [1]. Nevertheless, even in 90% of LBP cases, establishing an unambiguous cause of the symptoms is impossible and the condition is described as non-specific low-back pain (NLBP) [2].

So far, a contributory role of thoracolumbar fascia (TLF) dysfunction in some cases of LBP has been suggested. The TLF covers a substantial portion of muscles of the back and has significantly more sensory nerve endings than the underlying muscles [3,4]. Furthermore, it has been confirmed that patients with LBP also exhibit structural changes in the TLF that may lead to incorrect tensions that increase muscle dysfunction and generate pain, making it difficult to address conditions that have been present for a long time [5,6]. It is believed that two of the likely causes of NLBP are local myofascial disorders of the richly innervated TLF and distal disorders that impact nerves that supply the area where the pain is felt through numerous functional connections, thus inducing lumbosacral spine disorders. Some authors suggest defining a specific category for myofascial pain syndrome (MPS) that would be associated with microtrauma that irritates nociceptor nerve endings in the TLF [7], increased concentration of CGRP (calcitonin gene-related peptide) and SP (substance P) in an inflamed TLF [8], as well as reduction of sliding movements between TLF layers [9].

LBP is a condition associated with changes in muscle activity in the trunk. The above has been confirmed by surface electromyography, which is a simple and non-invasive diagnostic method for measuring electrical activity of selected muscle groups both in static conditions and during movement [10]. Resting activity of paraspinal muscles in the lumbosacral spine (erector spinae and superficial fibers of the multifidus) in patients with LBP in a relaxed standing position is significantly greater than that of people without the condition [11,12].

One treatment option for LBP is manual myofascial release techniques that aim to increase elasticity and sliding between layers of soft tissues, thus decreasing muscle activity and pain intensity as well as improving a patient’s functional fitness [13,14,15]. So far, there are no studies examining immediate changes in muscle activity following a single MFR session.

## 2. Study Objective

Assessment of changes in resting activity of selected muscles of the thoracolumbar fascia after a single MFR treatment in a group of patients with CLBP immediately after the intervention, and one month after the therapy. To achieve the aforementioned goals, the following research hypotheses were formed:

**Hypothesis** **1.**
*Resting activity of the erector spinae and multifidus muscles in a relaxed standing position will be lower immediately after the intervention and 1 month after the treatment compared to that recorded before the treatment.*


**Hypothesis** **2.**
*Change in muscle activity will be greater in the experimental group compared to the control group.*


## 3. Methods

### 3.1. Study Design

The study was conducted between June 2019 and March 2020. A total of 158 people with LBP volunteered to participate in the study. Subjects were enrolled using an original interview questionnaire and a functional exam according to the following criteria.

The inclusion criteria were:Age 40–60 years;Chronic low-back pain (lasting more than 3 months).

The exclusion criteria were:Neurological conditions, previous surgical treatment, spine injury, contraindications to MFR treatment (acute inflammations, viral and bacterial infections, infectious diseases, fever, deep vein thrombosis, active malignant disease, aneurysms), contraindications to surface electromyography (sEMG) exam (pacemaker, artificial heart valve);Physiotherapeutic interventions in the last 6 months;The following coexisting conditions—cancer, diabetes, osteoporosis, pregnancy; digestive system, cardiovascular system, rheumatic, psychic, and gynaecological diseases.

We excluded patients who exhibited neurological symptoms during a functional exam that consisted of:The slump test;The Babinski test and ankle clonus test;Sciatic nerve tests (straight leg raise and bowstring test), femoral nerve test (in the supine lying position and side-lying position);Knee and ankle reflexes test, strength testing of indicator muscles in the lumbosacral spine;Evaluation of exteroceptive sensation along dermatomes associated with lumbosacral nerve roots.

At this stage researchers excluded 39 subjects from the study: several people refused to participate in the study (*n* = 19) and others (*n* = 20) did not meet the study inclusion criteria or met the study exclusion criteria. Simple randomization was applied to assign 119 subjects to study groups. Subject allocation was done using a set of 119 sealed envelopes. Each envelope contained a sheet of paper with an even or odd number (1–119), and each enrolled participant was asked to draw one sealed envelope. Subjects with odd numbers (*n* = 60) were allocated to the experimental group (EG) that underwent MFR therapy. The remaining participants with even numbers (*n* = 59) were assigned to the control group (CG) and did not receive any therapeutic intervention. The EG was assessed three times—before the intervention, immediately after the intervention, and one month after the therapy, whereas the CG was assessed twice—before the intervention and one month after the study. The study participants were asked to refrain from any physiotherapy treatment in the month between the intervention and the control visit. At the assessment visit one month after the therapy, researchers excluded one subject from the EG (ankle injury) and 5 subjects from the CG (2 participants missed the control visit, and 3 people received physiotherapy for the spine within the last month). Ultimately, 113 subjects completed the study. Researchers used the Consolidated Standards of Reporting Trials (CONSORT) statement to improve the RCT reporting quality (Figure 1) [16]. The experimental group consisted of 27 women and 33 men, aged 41–60 (Me = 49.36; SD = 5.91). The control group consisted of 28 women and 26 men, aged 41–60 (Me = 48.91; SD = 5.38). Most respondents claimed that they perform or used to perform mainly white-collar work (EG = 72.27%; CG = 53.70%) in sedentary position (EG = 77.97%; CG = 53.70%).

### 3.2. Measurements

Surface electromyography was registered from the TLF to determine resting muscle activity level. For this purpose, the researchers used a 4-channel Noraxon MyoTrace 400 electromyograph, MyoResearch XP Clinical Edition 1.08 software, and disposable hypoallergenic Ag/AgCl gel electrodes. We used the equipment, prepared the exam, collected and rectified signals in accordance with the SENIAM guidelines [17]. During the study we recorded sEMG signals of the erector spinae (ES) and the multifidus muscle (MF) in the lumbosacral spine. Electrode placement is shown in Figure 2. The reference electrode was placed on the spinous process.

The study protocol consisted of 4 consecutive parts:Assessment of the resting activity in a relaxed standing position (10 s);Trunk flexion (forward bend) within available range of motion performed with knee extension (5 s);Maintaining trunk flexion with knee extension and relaxed shoulder girdle and upper limbs (5 s);Returning to a standing position—trunk extension (5 s).

Raw sEMG signals were rectified and then polished using a Root Mean Square algorithm (RMS). Since several measurements were conducted at time intervals and the results were compared between the groups, it was necessary to normalize the signal amplitude. Study participants were patients with LBP, and therefore it was not possible to establish maximum isometric tension for the examined muscles against static resistance (before the actual measurement), the amplitude of which is commonly used as a reference value among healthy and athletic individuals [18]. Using a protocol containing dynamic activities, including concentric muscle work during trunk extension (stage 4), made it possible to use a solution, according to which the maximum voluntary contraction (MVC) obtained from the actual measurement served as a reference value (Figure 3). MVC is defined as the arithmetic mean of the amplitude of the highest segment of the signal with a constant duration of 1000 ms. The recorded data was converted from microvolts to percentage of the reference value without altering the shape of the EMG curve [18,19].

Since the aim of this study was to assess changes in the resting activity of ES and MF muscles in a standing position, the statistical analysis was performed using mean values of %MVC recorded in the first stage of the study.

### 3.3. Intervention

The experimental group received a single session of MFR therapy. To ensure the highest quality of techniques, strokes and optimal tissue control, the researchers used MFR Songbird Fascial Release Wax which is a natural wax used in MFR. The techniques were applied according to the guidelines described by Luchau and engaged all three layers of TLF [13]. The sequence of the techniques, performed in accordance with principles laid out by Myers and Manheim, allowed us to address soft tissues gradually, moving from the most superficial layers to deeper structures [20,21]. The following techniques were applied—in the supine lying position: skin rolling [14], local relaxation of ES using cross-handed stretch [21], pelvic and spinal distraction [14]; and in the side-lying position: longitudinal stretching of the ES and the MF muscles in fetal position [14], hook and stretch of the posterior and middle layers of thoracolumbar fascia [15], release of the quadratus lumborum (direct stretching [14] and hook and stretch [15]). Each technique was performed on both sides, and the whole intervention lasted for 40 min.

### 3.4. Statistical Analyses

Reported data analyses, visualizations and necessary data operations were performed using R statistical environment, ver. 3.6.2. Operations on data were carried out using the data.table library, whereas the graphs were plotted using elementary R functions as well as ggplot2 and bayesplot libraries. To test the research hypotheses, hierarchical Bayesian regression was employed using the brms library [22]. The therapeutic effects in the EG were analyzed by matching the data to the null model. It should be noted that the null model presented merely a random effect to a patient. The next stage involved estimating parameters of the measurement model that contained effects coded 0/1 (immediately after the intervention and one month after therapy vs. before therapy). Greater predictive power of the measurement model compared to the null model suggests the validity of the research hypothesis regarding the effectiveness of the intervention in the EG. That is, of course, assuming that the observed effects are in the assumed direction. The researchers conducted a stepwise comparative analysis of the CG and the EG. In the first step, data were fitted to the null model, then the main effects model (the EG model) and finally to the main effects and interaction model (the EGI model). The EG model contained main effects for the group (coded 0 for the CG and 1 for the EG), and the measurement (coded 0 for pre-treatment and 1 for one month following the intervention). The EGI model contained main effects and the interaction effect between a group and the intervention. Best fit was obtained with the EGI model, which suggests that a change in the CG parameter is reliably different from the change in the EG parameter.

To compare the predictive power of the estimated regression models, the researchers used two information criteria calculated using the cross-validation method—the leave-one-out information criterion (LOOIC), and the k-fold information criterion (KFOLDIC). Both statistics measure the predictive power of the model outside the sample, i.e., the model’s ability to correctly predict new observations. Since the LOOIC and KFOLDIC values are based on logarithms of probabilities, differences between their values always indicate a strong evidence in favor of the better model. That is because differences between 110 and 100 and between 1010 and 1000 are equally strong evidence—10 units, irrespectively of the baseline statistics—in favor of the better-fitted model. LOOIC was the default statistic; however, in case of unreliable LOOIC approximation, KFOLDIC was used [23]. The interpretation of both statistics is simple–lower values indicate a better fit to the data. Similarly to other information criteria, differences greater than 3 (absolute) units constitute weak evidence in favor of a better model. A 4–7 difference constitutes medium-strong evidence, whereas a difference >10 is strong evidence in favor of the better model [24]. Moreover, to provide an intuitive measure of the effect’s magnitude, we reported R-squared for Bayesian for models using a continuous dependent variable [25].

## 4. Results

The research hypotheses were tested using the MVC variable proportion, which is a four-dimensional right-skewed continuous variable within the 0–1 range (Figure 4). For the purpose of improving the stability of the parameter estimation procedure, the %MVC was converted to proportions (decimal), that is the 0–1 range. Moreover, the researchers applied the lognormal probability distribution used in analysis of such variables.

To test the H1 research hypothesis that the values of the % MVC variable would be lower immediately after the intervention and one month after treatment compared to the result before treatment, the researchers performed hierarchical four-dimensional lognormal regression analyses. The proper model was better fitted to the data than the null model (LOOIC difference = −11), and the effect of time measurement accounted for approximately 2% of left MF variance, right MF variance, and left ES (Table 1). The measurement time for the right ES accounted for 5% of the %MVC variance, thus suggesting that the greatest effect was recorded for this muscle. As expected, reliably lower %MVC values were recorded for all muscles immediately after the intervention and one month after treatment, compared to the result measured before treatment (Figure 5).

To test the H2 research hypothesis that the change of the %MVC variable would be greater in the EG than the CG, the researchers also performed hierarchical four-dimensional lognormal regression analyses. The model with the interaction effect proved to be better fitted to the data than the model with the main effects (KFOLDIC difference = −21), and the null model (LOOIC difference = −15). Obtained results suggest there are reliable differences for %MVC between the study groups. Main effects as well as measurement time and group interaction accounted for 7% of the left MF variance, 6% of the right MF variance, 2% of the left ES, and 3% of the right ES (Table 2). As predicted, a reliably greater decrease in %MVC values for measurements for all muscles was recorded in the EG compared to the CG (Figure 6); however, these effects are more evident for the multifidus muscle.

## 5. Discussion

LBP is among the most common musculoskeletal problems affecting modern society worldwide. Complex multifactorial etiology of this condition significantly hinders the diagnosis of the primary cause. Therefore, numerous authors undertake research to understand better the complexity of LBP. Connective tissue problems have been dismissed as causes of LBP until recently, and in recent years there has been a growing interest in this matter. Currently, thanks to the groundbreaking results of Willard [26], Schleip et al. [27,28] and Stecco et al. [9,29,30] we know more about the physiology of the connective tissue and understand better its crucial role in proper functioning of the motor system. For this reason, too, the researchers continue looking for more effective methods of conservative treatment that bring quicker and longer-lasting therapeutic effects, while analyzing their impact on the myofascial system. The analysis of the results of studies evaluating the effects of MFR therapy in patients with LBP revealed that most researchers used original and standardized interview questionnaires, in which the respondents subjectively rated their pain intensity before and after the intervention. According to our current knowledge, no studies investigated immediate changes in muscle activity in patients with LBP following a single MFR session. Therefore, the authors decided to examine this study gap and objectively measure the parameters using sEMG. Our study findings demonstrate that there is a statistically reliable reduction in muscle activity within TLF following a single MFR treatment. Therapeutic results are present immediately after the intervention. Furthermore, measurements carried out one month after the intervention confirm that therapeutic effects are maintained because with time, values of the above parameters did not revert to the baseline values obtained before the treatment. While a reliable reduction in the resting activity of ES and MF muscles was observed in the EG, the researchers also noted an increase of this parameter in the CG after one month. Even though reliable differences were observed in the study groups for the activity of both muscles, greater differences were found in the MF activity. Our literature review yielded that there are no studies that would analyze the impact of a single MFR session on the immediate change in resting muscle activity in the TLF. Moreover, reliability of an immediate impact of MFR on patients with LBP was confirmed several times by authors who used other objective research tools. The findings in those studies confirm that even a single MFR session significantly increases lumbar spine range of motion [31], improves sliding movements between TLF layers [32,33], and, according to Shah et al., improves lumbar paraspinal blood flow [34]. A paper by Arguisuelas et al. examines myoelectric activity before and after several sessions of myofascial treatment [35]. Those authors, using a 4-step study protocol identical to the one outlined in this paper, analyzed only the activity of ES in forward bend position, which should be significantly reduced due to the flexion–relaxation phenomenon (FRP). Following 4 sessions of MFR in the TLF there was a bilateral normalization of the ES muscle activity and an improvement of FRP in participants who did not exhibit a considerable decrease in myoelectric activity during full trunk flexion before the intervention. Given the fact that there are multiple myofascial connections throughout the body, designing a single myofascial release standard protocol for patients with LBP seems highly challenging. However, based on positive outcomes following a single and several sessions of MFR, we believe that this topic needs further investigation and our intention is to undertake it in the future.

## 6. Conclusions

A single MFR treatment in a group of patients with CLBP immediately decreases resting activity of the erector spinae and multifidus muscles in the lumbodorsal spine area. The comparison of the results with those of the control group results reveal that the effects are definitely stronger for the multifidus muscle.Data collected a month after the treatment confirm maintenance of the treatment effect in terms of muscular activity of the erector spinae and multifidus muscles in the lumbosacral spine.

## 7. Limitation

The authors acknowledge certain limitations of their analysis. These include the lack of use of sham therapy in the control group and the limited ability to control the study participants in the period between measurements, especially in the psycho-emotional aspect, which could have a potential impact on resting muscle activity.

## Figures and Tables

**Figure 1 jcm-10-04039-f001:**
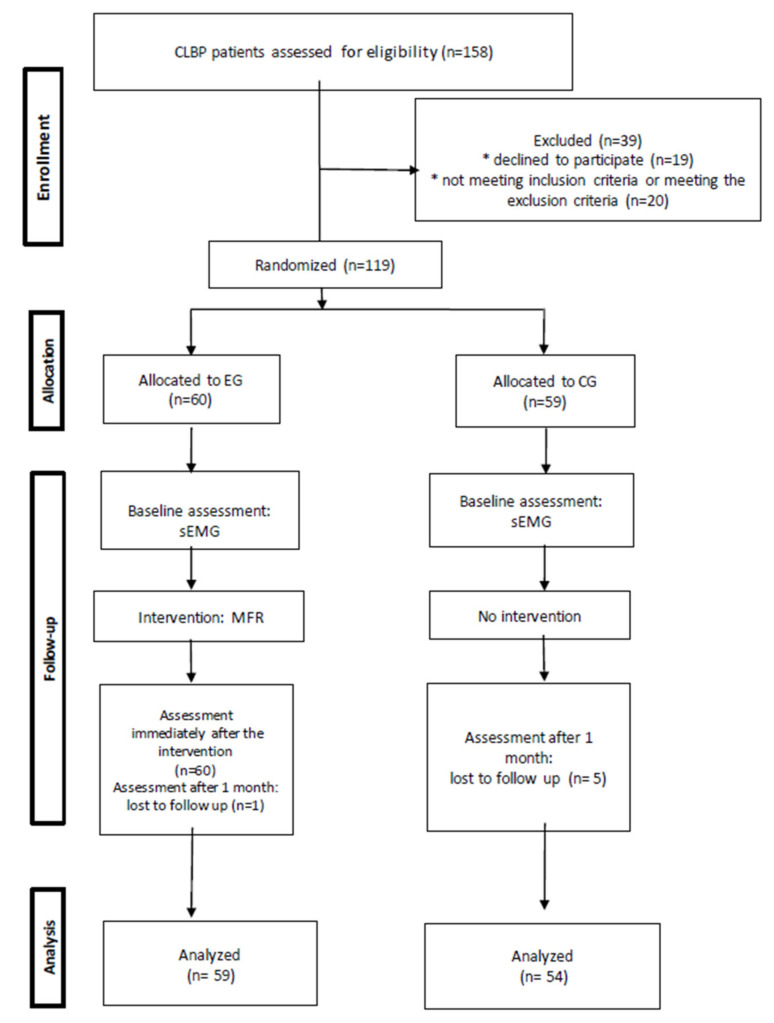
The study flow diagram. CLBP—chronic low-back pain, EG—experimental group, CG—control group, sEMG—surface electromyography, MFR—myofascial release.

**Figure 2 jcm-10-04039-f002:**
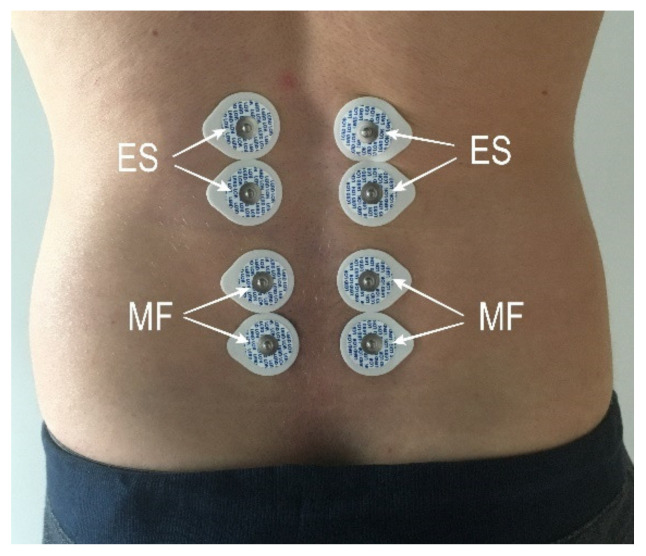
Arrangement of surface electrodes.

**Figure 3 jcm-10-04039-f003:**
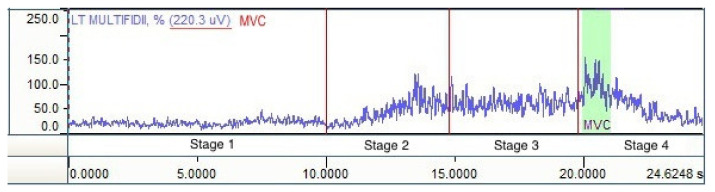
sEMG record [%MVC] following signal amplitude normalization. Author’s content.

**Figure 4 jcm-10-04039-f004:**
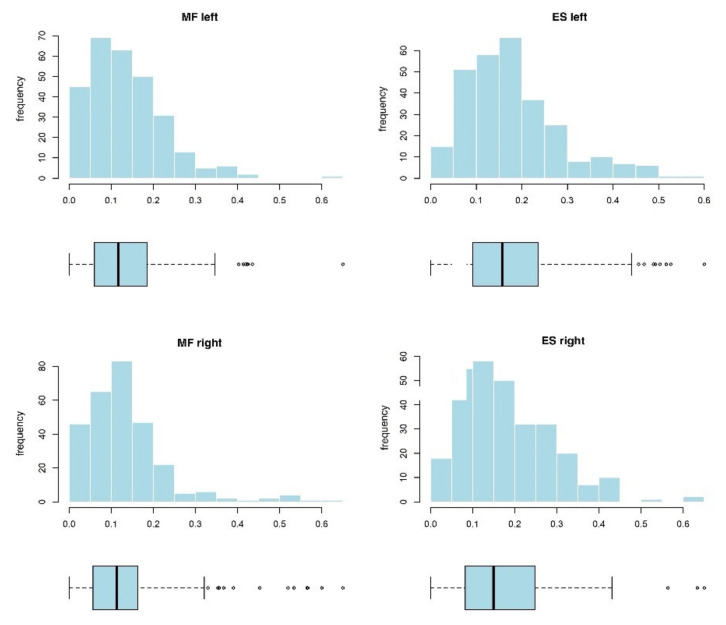
Histograms and box plots for the two-dimensional MVC variable. The graphs are based on all the data collected from the EG and the CG.

**Figure 5 jcm-10-04039-f005:**
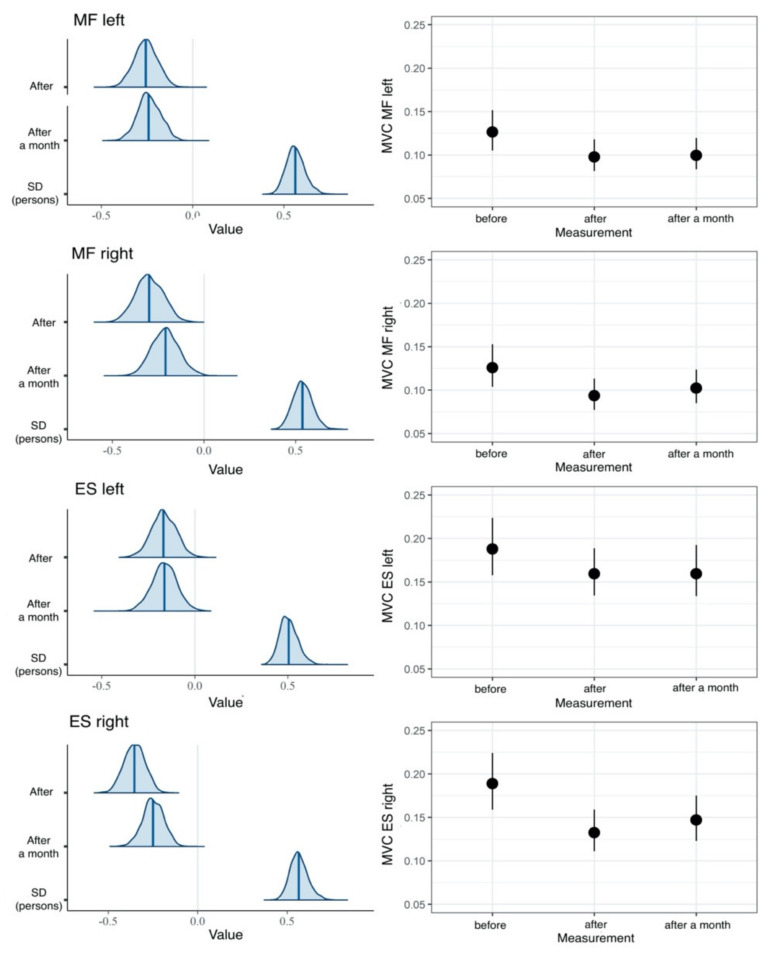
Results of hierarchical four-dimensional lognormal regression analysis with MVC as the dependent variable and the measurement as the main effect in the study group. Left panels: a posteriori distributions of regression coefficients. The shaded areas represent 95% confidence areas. Right panels: estimated marginal means. The points represent means of a posteriori distributions, whereas vertical lines indicate 95% confidence intervals.

**Figure 6 jcm-10-04039-f006:**
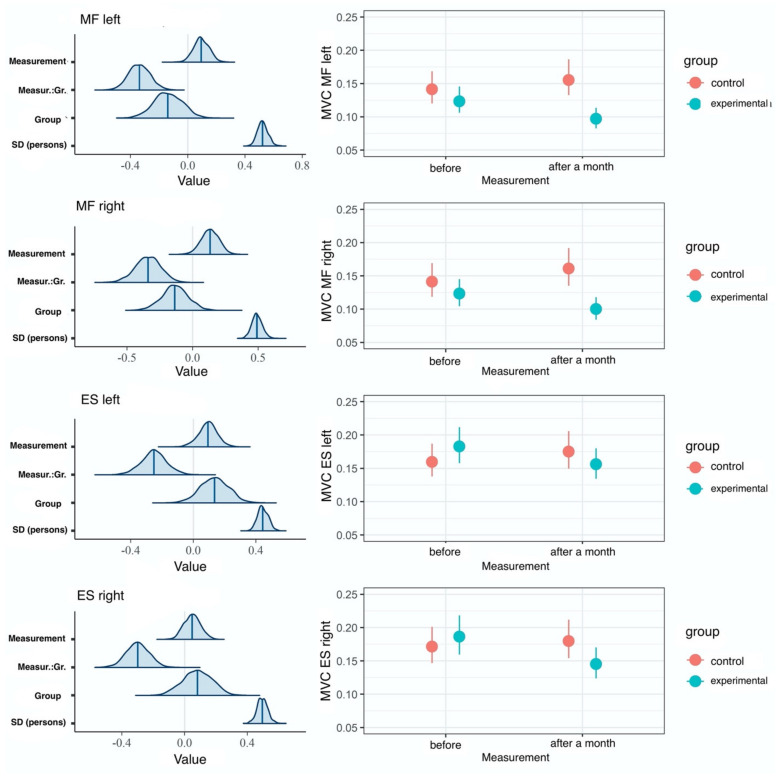
Results of hierarchical four-dimensional lognormal regression analysis with MVC as the dependent variable, and the measurement and the group as predictors. Left panels: a posteriori distributions of regression coefficients. The shaded areas represent 95% confidence areas. Right panels: estimated marginal means. The points represent means of a posteriori distributions, whereas vertical lines indicate 95% confidence intervals.

**Table 1 jcm-10-04039-t001:** Summary of regression models’ fitness to the data used for testing the research Hypothesis 1.

Dependent Variable	Model	LOOIC/KFOLDIC	Dependent Variable 2	R-Squared EG	Total R-Squared
MVC	null model	−1945	MF left	-	-
			MF right		
			ES left		
			ES right		
	measurement	−1956	MF left	0.02 [0.01, 0.05]	0.74 [0.64, 0.80]
			MF right	0.02 [0, 0.05]	0.58 [0.43, 0.69]
			ES left	0.02 [0, 0.05]	0.62 [0.52, 0.70]
			ES right	0.05 [0.02, 0.09]	0.69, [0.62, 0.74]

LOOIC—the leave-one-out information criterion, KFOLDIC—k-fold information criterion, MVC—maximum voluntary contraction, MF—multifidus muscle, R-squared EG—R-squared for the measurement effects and a group. Total R-squared is a variance explained by a random effect to a study participant.

**Table 2 jcm-10-04039-t002:** Summary of regression models’ fitness to the data used for testing the research Hypothesis 2.

Dependent Variable	Model	LOOIC/KFOLDIC	Dependent Variable 2	R-Squared EG	total R-Squared
MVC (kfold)	null model	−1865	MF left	-	-
			MF right		
			ES left		
			ES right		
	EG	−1859	MF left	0.06 [0.01, 0.13]	0.73 [0.63, 0.80]
			MF right	0.04 [0, 0.11]	0.58 [0.43, 0.70]
			ES left	0.01 [0, 0.04]	0.63 [0.52, 0.72]
			ES right	0.02 [0, 0.06]	0.67 [0.58, 0.73]
	EGI	−1880	MF left	0.07 [0.02, 0.14]	0.75 [0.67, 0.82]
			MF right	0.06 [0.02, 0.12]	0.60 [0.46, 0.72]
			ES left	0.02 [0, 0.06]	0.64 [0.53, 0.73]
			ES right	0.03 [0.01, 0.07]	0.68 [0.6, 0.74]

LOOIC—the leave-one-out information criterion, KFOLDIC—k-fold information criterion, MVC—maximum voluntary contraction, MF—multifidus muscle, R-squared EG—R-squared for the measurement effects and a group. Total R-squared is a variance explained by a random effect to a study participant.

## Data Availability

The data sets generated during and/or analyzed during the current study are available from the corresponding author on reasonable request.

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
