# Peer review of "Analysis of Muscle Activity Following the Application of Myofascial Release Techniques for Low-Back Pain—A Randomized-Controlled Trial"

_jcm, 2021, doi:10.3390/jcm10184039_

Round 1
Reviewer 1 Report
Comments to the author
General comments
Although the results provide data that may be interesting, it would be necessary to include key methodological and data that would make the study robust. The authors justify the study design, the evaluation model, the robustness of the results, the interpretation of the results obtained are easy to generalize.
However, it is recommended to include a series of key methodological data to improve understanding of the results and their clinical applicability.
Specific comments
Abstract
- The introduction is too long for an abstract. It can be summarized more.
- The abbreviations ES and MF appear before the full name. Review.
- The type of study (study design) is not stated.
- The authors should indicate some characteristic of the variables measured with the EMGs since this instrument measures several variables.
- No characteristics of the intervention carried out are indicated. A minimal information would be interesting for future readers.
- No statistical value is indicated in the Results
- The conclusions are very strong. However, based on the indicated results, there may be doubts. The best option would be to provide more information in the results.
Introduction
- The justification for the intervention carried out with the clinic and the characteristics of the low back pain can be improved. There is hardly any impact on the reason for this intervention.
- It would be of interest to the reader, what types of exercises are continually referred to in the Introduction section.
- Myofascial release is listed at the end of the section. Its efficacy in improving cervical pain and range of motion should be indicated in previous studies.
Methods
- Study design is not indicated.
- Was the person in charge of managing the randomization process and assigning patients to the study groups blinded?
- The recruitment dates, the mode of inclusion of the patients (from where she contacted them), locations or intervention dates are not indicated.
- The data of approval of the ethics committee or registration of the study in a national or international database are not indicated.
- To facilitate the understanding and replicability of the interventions, the authors should indicate, in detail, all the techniques used in the experimental group. This information is essential and is not clear in the text. The inclusion of a figure would help the reader who is not familiar with myofascial techniques.
- Was the evaluator blinded?
- The Statistical Analysis section is too complex to follow. For readers unfamiliar with the handling of R, I advise that the authors follow a simpler scheme: test used and its objective to confirm each objective established in the study.
Results
- A table with the values of central tendency and dispersion in both groups, in all evaluations, could be of interest for statistical analysis of future studies / meta-analyzes.
- The simplification of the statistical values (significant changes and the calculation of the effect size) would further facilitate the understanding of the results and their clinical importance.
Discussion
- The authors should justify in a more organized and structured way the reason for the changes observed in the study.
- How did the authors control that in the month of “follow-up” there were no variables that could influence the results? Achieving effects with a single intervention is not impossible (in clinic it is common), but its maintenance for one month and its improvement compared to a control group, can make clinicians doubt. This aspect should be very well justified, because it can be key in the results and conclusions of the study.
- A Limitations study and Relevance for clinical practice sections should be included in the text.
Conclusions
- In the absence of results that confirm the statistical power, perhaps the authors are very categorical in their conclusions, with a single intervention, without controlling for other variables that could influence.
Author Response
Dear Reviewer,
I am writing to express our sincere gratitude to you for having reviewed our manuscript on “Analysis of muscle activity following the application of myofascial release techniques for low back pain – a randomized-controlled trial”
We also greatly appreciate all the constructive comments provided by the reviewers as they helped us to improve the quality of this manuscript. We revised the article and implemented several changes.
In the text below, you will find detailed answers to reviewers’ comments.
Reviewer 1
The introduction is too long for an abstract. It can be summarized more.
We would like to keep all information as we consider it important.
The abbreviations ES and MF appear before the full name. Review.
Corrected
The type of study (study design) is not stated.
Added in the title
The authors should indicate some characteristic of the variables measured with the EMGs since this instrument measures several variables.
We have given the exact name of the EMG device, but from the wide range of possibilities of the device, we have discussed only those used in the study
No characteristics of the intervention carried out are indicated. A minimal information would be interesting for future readers.
Described in 3.3. Intervention
No statistical value is indicated in the Results
The conclusions are very strong. However, based on the indicated results, there may be doubts.
According to the assumptions of Bayesian statistics, the EGI model contained main effects and the interaction effect between a group and the intervention. Best fit was obtained with the EGI model which suggests that a change in the CG parameter is reliably different than the change in the EG parameter. In accordance with the assumptions of the Bayesian statistics presented in 3.4, to compare the predictive power of the estimated regression models, the researchers used two information criteria: LOOIC and KFOLDIC. Differences between their values always indicate a strong evidence in favour of the better model. Differences greater than 3 (absolute) units constitute weak evidence in favour of a better model. A 4–7 difference constitutes medium-strong evidence, whereas a difference >10 is strong evidence in favour of the better model. In our work, in comparisons made within the EG, as well as between groups, differences between values of information criteria showed strong evidence (Table 1 and Table 2), which confirms statistically reliable differences between the groups. Therefore, in our opinion the conclusions are based on the research results.
The best option would be to provide more information in the results.
In this study, we only wanted to show these results.
The justification for the intervention carried out with the clinic and the characteristics of the low back pain can be improved. There is hardly any impact on the reason for this intervention.
This intervention is a relatively new therapy, hence our interest in its effectiveness. At the end of the introduction, citing information provided by other authors, information about the potential effects of MFR application in LBP patients was included. Some of these effects have been objectively confirmed in the studies of other authors presented in the discussion of this paper [31-34], which confirmed us in the rightness of applying this type of intervention.
Myofascial release is listed at the end of the section. Its efficacy in improving cervical pain and range of motion should be indicated in previous studies.
Described in the discussion. Due to the subject of the work, including the lumbosacral spine, studies by other authors who used MFR in this specific area were quoted.
Study design is not indicated.
The study design was described in great detail ( Fig. 1. The study flow diagram )
Was the person in charge of managing the randomization process and assigning patients to the study groups blinded?
Yes, this person was blind
The recruitment dates, the mode of inclusion of the patients (from where she contacted them), locations or intervention dates are not indicated.
added in 3.1. Study Design
The data of approval of the ethics committee or registration of the study in a national or international database are not indicated.
Indicated (at the end of the manuscript):
Institutional Review Board Statement: The study was conducted in accordance with the Declaration of Helsinki guidelines. Prior to the study, the authors obtained approval from the Bioethics Committee of the Collegium Medicum in Bydgoszcz, Nicolaus Copernicus University in Torun (KB: 588/2017).
To facilitate the understanding and replicability of the interventions, the authors should indicate, in detail, all the techniques used in the experimental group. This information is essential and is not clear in the text. The inclusion of a figure would help the reader who is not familiar with myofascial techniques
We assumed that we will give general information, because a detailed description of this technique for the purposes of this manuscript would be very difficult. If the reader would like to get acquainted with the techniques used in the work in more detail, the text contains references to the literature describing each of the techniques.
Was the evaluator blinded?
No, he is not blinded
The Statistical Analysis section is too complex to follow. For readers unfamiliar with the handling of R, I advise that the authors follow a simpler scheme: test used and its objective to confirm each objective established in the study.
We would prefer to stick to the Bayesian statistics. In our opinion, the assumptions of this statistical approach require a short description for better understanding and readability of the results section of the work.
Table with the values of central tendency and dispersion in both groups, in all evaluations, could be of interest for statistical analysis of future studies / meta-analyzes.
For the purposes of meta-analyzes, we can provide the results
The simplification of the statistical values (significant changes and the calculation of the effect size) would further facilitate the understanding of the results and their clinical importance.
We would prefer to stick to the Bayesian statistics. The method of determining the significance of the obtained results is clearly presented in section 3.4, and the specific values are presented in the tables (Table 1 and Table 2).
How did the authors control that in the month of “follow-up” there were no variables that could influence the results? Achieving effects with a single intervention is not impossible (in clinic it is common), but its maintenance for one month and its improvement compared to a control group, can make clinicians doubt. This aspect should be very well justified, because it can be key in the results and conclusions of the study.
We believe that this is what scientific research is for to confirm or contradict an assumed research hypothesis. We agree with the Reviewer that it is difficult to control that in the month of "follow-up" there were no variables that could influence the results. However, to conduct this study, we had to trust our patients - they were informed about the research assumptions at the recruitment stage. Nevertheless, in line with this important and valid remark, we added a section on the limitations of our study at the end of the manuscript, which also included the limited possibility of full control over patients between measurements - especially in the psycho-emotional aspect, which could have a potential impact on resting muscle activity.
Reviewer 2 Report
I congratulate the authors for this well orchestrated investigation and also the excellent manuscript.
The following are suggestions for possible improvements. They are not to be confused with "requirements"; and if after careful consideration you decide to keep the respective manuscript formulations unchanged, based on your in-depth background of the background behind, that will be fine.
Line 10-11: "... the leading role ... in causing LBP has been demonstrated". I suggest that this is a misleading statement; or a premature conclusion based on the available scientific data. To my extensive knowledge of the current research on fascia related changes in LBP, these studies do not yet show that the described fascial changes have a causal role in LBP; only that they are are associated with LBP. It seems more than likely to me, based on the current scientific data, that fascial factors do "probably" play a causal or contributory role in "some cases of LBP" today. But whether that is "the leading cause", more than any non-fascial factor, will be a matter of future investigations. How about a more modest statement like "A contributory role of thoracolumbar (TLF) dysfunction in some cases of LBP has been suggested"?
L37: same remark as above
L8-26: It may be valuable to include the fact that the patient allocation was randomized into the abstract. Without such randomization the quality of the study would be much lower than it current is.
L164-166. Not sure if the "anterior layer" in the three-layer model described in Willard et al. (doi: 10.1111/j.1469-7580.2012.01511.x.) is included in that statement. In case it is not included, then maybe a good solution could be ".. all the layers of the TLF, as described in the two-layered model of Willard et al."
L271-273. Are you completely sure about this bold statement? If yes, then you can keep it of course. If not, maybe a more modest formulation would be better, such as "to our current knowledge" or similar. In particular check this quick finding from me today: DOI: 10.1016/j.jbmt.2021.03.015 (published online in March 2021).
L283-285: this sentence would need be specified to "resting muscle activity" in order to be more correct. But then it seems redundant with L271-273. Please ask yourself whether such a repetition is required.
L286: Maybe more clear to write "The findings in those studies confirm ..." Otherwise one might get the wrong impression that you are describing the results of your own study.
L305. Maybe better "in the lumbodorsal spine area"?
In general: Your investigation did not include comparison with a sham treatment (such as sham-laser treatment of sham-ultrasound treatment for LBP) with associated blinding of patients and evaluators. It is therefore quite possible, that the psycho-emotional state of the patients in the treatment group was different in the measurement situation as compared with the control group; would may have influenced their resting muscle EMG in standing. I suggest to reflect about this small potential corroborating factor with the 'Discussion' section of this paper. At least it needs to be briefly mentioned as a potential (but small) weakness of the chosen methodological design.
Author Response
Dear Reviewer,
I am writing to express our sincere gratitude to you for having reviewed our manuscript on “Analysis of muscle activity following the application of myofascial release techniques for low back pain – a randomized-controlled trial”
We also greatly appreciate all the constructive comments provided by the reviewers as they helped us to improve the quality of this manuscript. We revised the article and implemented several changes.
In the text below, you will find detailed answers to reviewers’ comments.
Reviewer 2
Line 10-11: "... the leading role ... in causing LBP has been demonstrated". I suggest that this is a misleading statement; or a premature conclusion based on the available scientific data. To my extensive knowledge of the current research on fascia related changes in LBP, these studies do not yet show that the described fascial changes have a causal role in LBP; only that they are are associated with LBP. It seems more than likely to me, based on the current scientific data, that fascial factors do "probably" play a causal or contributory role in "some cases of LBP" today. But whether that is "the leading cause", more than any non-fascial factor, will be a matter of future investigations. How about a more modest statement like "A contributory role of thoracolumbar (TLF) dysfunction in some cases of LBP has been suggested"?
L37: same remark as above
Of course, we fully agree with the proposed remark. In fact, research indicates the occurrence of structural changes in the TLF area among people with LBP, which, however, absolutely does not confirm that they would be the primary cause of these ailments. We meant exactly that, but used a wording that was too bold. We have made the proposed changes to the text.
L8-26: It may be valuable to include the fact that the patient allocation was randomized into the abstract. Without such randomization the quality of the study would be much lower than it current is.
Information on randomization has been added to the abstract and the title of the paper.
L164-166. Not sure if the "anterior layer" in the three-layer model described in Willard et al. (doi: 10.1111/j.1469-7580.2012.01511.x.) is included in that statement. In case it is not included, then maybe a good solution could be ".. all the layers of the TLF, as described in the two-layered model of Willard et al."
Yes, the guidelines of Til Luchau, quoted in this sentence, for the manual development of TLFs specifically included the three-layer model, so we clarified that the techniques used included the three-layer TLFs.
L271-273. Are you completely sure about this bold statement? If yes, then you can keep it of course. If not, maybe a more modest formulation would be better, such as "to our current knowledge" or similar. In particular check this quick finding from me today: DOI: 10.1016/j.jbmt.2021.03.015 (published online in March 2021).
While we have carefully searched the databases for this type of study before making this claim, we agree that it is safer to use a more modest wording as suggested.
L283-285: this sentence would need be specified to "resting muscle activity" in order to be more correct. But then it seems redundant with L271-273. Please ask yourself whether such a repetition is required.
Corrected 283-285. L271-273 remained unchanged
L286: Maybe more clear to write "The findings in those studies confirm ..." Otherwise one might get the wrong impression that you are describing the results of your own study.
That's right - we're referring to the results of other authors here. Corrected.
L305. Maybe better "in the lumbodorsal spine area"?
Corrected.
In general: Your investigation did not include comparison with a sham treatment (such as sham-laser treatment of sham-ultrasound treatment for LBP) with associated blinding of patients and evaluators. It is therefore quite possible, that the psycho-emotional state of the patients in the treatment group was different in the measurement situation as compared with the control group; would may have influenced their resting muscle EMG in standing. I suggest to reflect about this small potential corroborating factor with the 'Discussion' section of this paper. At least it needs to be briefly mentioned as a potential (but small) weakness of the chosen methodological design.
We added a section on the limitations of our study at the end of the manuscript, which also included the limited possibility of full control over patients between measurements - especially in the psycho-emotional aspect, which could have a potential impact on resting muscle activity. Of course, we also added in this section that no sham treatment was used in the control group.